# VQ-TR: Vector Quantized Attention for Time Series Forecasting

## Abstract

Modern time series datasets can easily contain hundreds or thousands of temporal time points, however, Transformer based models scale poorly to the size of the sequence length constraining their context size in the seq-to-seq setting. In this work, we introduce VQ-TR which maps large sequences to a discrete set of latent representations as part of the Attention module. This allows us to attend over larger context windows with linear complexity with respect to the sequence length. We compare this method with other competitive deep learning and classical univariate probabilistic models and highlight its performance using both probabilistic and point forecasting metrics on a variety of open datasets from different domains.

## 1 Introduction

Deep learning based probabilistic time series forecasting models (Benidis et al., 2022) typically consists of a component that learns representations of a time series' history, while another component learns some emission model (point or probabilistic) conditioned on this representation of the history. One can typically use Recurrent Neural Networks (RNNs), Casual Convolutional networks or the popular architecture at time of writing namely, Transformers (Vaswani et al., 2017) to learn historical representations. Transformers offer good inductive bias for the forecasting task (Zhou et al., 2021), as they can look back over the full context history of a time series, while RNNs suffer from forgetting and convolutions network have limited temporal receptive fields for the amount of parameters. Transformers however suffer from quadratic complexity for memory and compute in the size of sequence over which they are learning representations. This constrains the size of contexts over which Transformer based models can make predictions over which can potentially hinder these models from learning long-range dependencies as well as constraining the depth of these models leading to poorer representations being learnt. A number of techniques have been developed to deal with this issue by reducing the computation or reducing the sequence length.

In this work we start with an observation on the approximation of the Attention module in the Transformer and design a novel encoder-decoder Transformer based architecture for the forecasting use case which is linear in its computational and memory use with respect to the sequence size. We do this by incorporating a Vector Quantization (van den Oord et al., 2017) into the Transformer which allows us to learn discrete representations in a hierarchical fashion.

## 2 Background

### 2.1 Probabilistic Time Series Forecasting

The task of probabilistic time series forecasting in the *univariate* setting consists of training on a dataset of $D \geq 1$ time series $\mathcal{D}_{\text{train}} = \{x^i_{1:T^i}\}$ where $i \in \{1, \ldots, D\}$ and at each time point $t$ we have $x^i_t \in \mathbb{R}$ or in $\mathbb{N}$. We are tasked with predicting the potentially complex distribution of the next $P > 1$ time steps into the future and we are given a back-testing test set $\mathcal{D}_{\text{test}} = \{x^i_{T^i+1:T^i+P}\}$. Each time index $t$ is in reality a date-time value which increments regularly based on the frequency of the dataset in question and the last training point $T^i$ for each time series may or may not be the same date-time. Autoregressive models like (Graves, 2013; Salinas et al., 2019b) estimate the prediction density by decomposing the joint distribution of all $P$ points via the chain-rule of probability as:

$$p_{\mathcal{X}}(x^i_{T^i+1:T^i+P}) \approx \Pi^P_{t=1} p(x^i_{T^i+t}|x^i_{1:T^i-1+t}, \mathbf{c}^i_{1:T^i+P}; \theta),$$

parameterized by some model with trained weights $\theta$. This requires that the next time point is conditioned on *all* the past and covariates $\mathbf{c}_t^i$ (detailed in Section 3.3), which is computationally challenging to scale especially if the time series has a large history. Models like `DeepAR` (Salinas et al., 2019b) typically resort to the seq-to-seq paradigm (Sutskever et al., 2014) and consider some context window of *fixed sized $C$* sampled randomly from the full time series history to learn some historical representation and use this representation in the decoder to learn the distribution of the subsequent time points of the context. This does however mean that the model falls short of capturing long-range temporal dependencies in its prediction which can lead to worse approximation of the future distribution.

Encoder-decoder Transformers (Vaswani et al., 2017) naturally fit the seq-to-seq paradigm where $N$ encoding Transformer layers can be used to learn a context window sized sequence of representations denoted by:

$$\{\mathbf{h}_t\}_{t=1}^{C-1} = \text{Enc} \circ \cdots \circ \text{Enc}(\{\texttt{concat}(x_t^i, \mathbf{c}_{t+1}^i)\}_{t=1}^{C-1}; \theta).$$

Afterwards $M$ layers of a *causal* or masked decoding Transformer can be used to model the subsequent $P$ future time points conditioned on the encoding representations as:

$$\Pi_{t=C}^{C+P-1} p(x_{t+1}^i | x_{t:C}^i, \mathbf{c}_{t+1:C+1}^i, \mathbf{h}_1, \ldots, \mathbf{h}_{C-1}; \theta).$$

For example if we assume the data comes from a Gaussian then the outputs from the Transformer's $M$ decoders can be passed to a layer which returns appropriately signed parameters of a Gaussian whose log-likelihood, given by

$$\sum_{t=C}^{C+P-1} \log p_{\mathcal{N}}(x_{t+1}^i | x_{t:C}^i, \mathbf{c}_{t+1:C+1}^i, \mathbf{h}_1, \ldots, \mathbf{h}_{C-1}; \theta),$$

can be maximized for all $i$ and $t$ from the $\mathcal{D}_{\text{train}}$ using stochastic gradient descent (SGD) as detailed in Section 3.1.

Although Transformers offer a better alternative to recurrent neural networks (RNN) (like the LSTM (Hochreiter & Schmidhuber, 1997) or GRU (Chung et al., 2014) which apart from being sequential suffer from forgetting with large context windows) or Convolutional models like `TCN` (Bai et al., 2018) (which have limited temporal receptive fields) they scale, in terms of compute and memory, quadratically with the size of sequence length *per* layer. To reduce the computational requirements of Transformers, which is an active area of research, one can employ a number of strategies, for example by compressing the sequence, exploiting locality or by mitigating computation for each of the input entity.

## 2.2 VECTOR QUANTIZATION (VQ)

The VQ-VAE (van den Oord et al., 2017; Razavi et al., 2019) is an encoder-decoder Variational Autoencoder (VAE) (Kingma & Welling, 2019) that maps inputs onto a set of $J \geq 1$ discrete latent variables called the codebook $\{\mathbf{z}_1, \ldots, \mathbf{z}_J\}$, and a decoder that reconstructs the inputs from the resulting discrete vectors. The input vector is quantized with respect to its distance to its nearest codebook vector:

$$\text{Quantize}(\mathbf{q}) := \mathbf{z}_n, n \quad \text{where} \quad n = \arg\min_j \|\mathbf{q} - \mathbf{z}_j\|_2. \tag{1}$$

The codebook is learned by back-propgation of the gradient coming upstream of the VQ module and due to the non-differentiable operation one uses the Straight-Through gradient estimator (Hinton et al., 2012; Bengio et al., 2013) to copy the gradients downstream. Additionally the VQ has two extra losses namely the latent loss which encourages the alignment of the codebook vectors to the inputs of the VQ as well as a commitment loss which penalizes the inputs from switching codebook vectors too frequently. This is done via the "stop-gradient" or "detach" operators of deep learning frameworks which blocks gradients from flowing into its argument. Thus the additional two VQ losses can be written as:

$$\|\texttt{sg}(\mathbf{q}) - \mathbf{z}\|_2^2 + \beta \|\texttt{sg}(\mathbf{z}) - \mathbf{q}\|_2^2, \tag{2}$$

where $\beta$ is the hyperparameter weighting the commitment loss. Since the optimal codes would be the k-means clusters of the input representations, van den Oord et al. (2017) provides an exponential

moving average training scheme of the latents instead of the latent loss (first term of (2)). Additionally, to aid learning, `Jukebox` (Dhariwal et al., 2020) proposes to replace the codebook vectors that have an exponential moving average cluster size less than some threshold by a random incoming vector from the batch.

## 3 `VQ-TR` MODEL

We motivate this method with an observation on the effect of approximations of the query vector in self-attention. Recall that in self-attention the incoming sequence of vectors are mapped to query, key, and value vectors. For each $t$ indexed vector, this is denoted by $\mathbf{q}_t$, $\mathbf{k}_t$, and $\mathbf{v}_t$, respectively. Let us denote the approximation of the query vector $\mathbf{q}_t$ by $\hat{\mathbf{q}}_t$. The attention weight for step $t$ attending on some $u$ step is (Phuong & Hutter, 2022)

$$w_{tu} = \frac{\exp(\mathbf{q}_t^T \mathbf{k}_u)}{\sum_j \exp(\mathbf{q}_t^T \mathbf{k}_j)},$$

and the output representation is denoted by $\mathbf{o}_t$, where $\mathbf{o}_t = \sum_u w_{tu} \mathbf{v}_u$. We then have the following:

**Theorem 1.** *If* $\max \left| \mathbf{q}_t^T \mathbf{k}_u - \hat{\mathbf{q}}_t^T \mathbf{k}_u \right| \leq \delta$ *then for sufficiently small* $\delta > 0$ *the attention weight with respect to the approximation* $\hat{\mathbf{q}}_t$ *given by* $\hat{w}_{tu}$ *is bounded by*

$$w_{tu}(1 - 2\delta) \leq \hat{w}_{tu} \leq w_{tu}(1 + 2\delta),$$

*and as a result the output representation is*

$$|\mathbf{o}_t - \hat{\mathbf{o}}_t| \preceq 2\delta \mathbf{o}_t.$$

*Proof.*

$$
\begin{aligned}
w_{tu} &= \frac{\exp(\mathbf{q}_t^T \mathbf{k}_u)}{\sum_j \exp(\mathbf{q}_t^T \mathbf{k}_j)} \\
&= \frac{\exp(\mathbf{q}_t^T \mathbf{k}_u - \hat{\mathbf{q}}_t^T \mathbf{k}_u + \hat{\mathbf{q}}_t^T \mathbf{k}_u)}{\sum_j \exp(\mathbf{q}_t^T \mathbf{k}_j - \hat{\mathbf{q}}_t^T \mathbf{k}_j + \hat{\mathbf{q}}_t^T \mathbf{k}_j)} \\
&= \frac{\exp(\hat{\mathbf{q}}_t^T \mathbf{k}_u) \exp(\mathbf{q}_t^T \mathbf{k}_u - \hat{\mathbf{q}}_t^T \mathbf{k}_u)}{\sum_j \exp(\hat{\mathbf{q}}_t^T \mathbf{k}_j) \exp(\mathbf{q}_t^T \mathbf{k}_j - \hat{\mathbf{q}}_t^T \mathbf{k}_j)}.
\end{aligned}
$$

Since, $\max_j \left| \mathbf{q}_t^T \mathbf{k}_j - \hat{\mathbf{q}}_t^T \mathbf{k}_j \right| \leq \delta$, then $\exp(-\delta) \leq \exp(\mathbf{q}_t^T \mathbf{k}_j - \hat{\mathbf{q}}_t^T \mathbf{k}_j) \leq \exp(\delta) \ \forall j$

$$\exp(-2\delta) \leq \frac{w_{tu}}{\hat{w}_{tu}} \leq \exp(2\delta)$$

assuming $\delta$ is small,

$$w_{tu}(1 - 2\delta) \leq \hat{w}_{tu} \leq w_{tu}(1 + 2\delta)$$

or,

$$|\hat{w}_{tu} - w_{tu}| \leq 2\delta w_{tu}.$$

Since $\mathbf{o}_t = \sum_u w_{tu} \mathbf{v}_u$,

$$|\mathbf{o}_t - \hat{\mathbf{o}}_t| \preceq \left| \sum_u w_{tu} \mathbf{v}_u - \hat{w}_{tu} \mathbf{v}_u \right| \preceq \sum_u |w_{tu} - \hat{w}_{tu}| \mathbf{v}_u \preceq \sum_u 2\delta w_{tu} \mathbf{v}_u \preceq 2\delta \mathbf{o}_t.$$

$\square$

With the above result we see that we can bound the error in the output representation of self-attention as a result of approximating the query vector. We can see how to make sure $\delta$ is small using a discrete set of approximations given by $\{\mathbf{z}_1, \ldots, \mathbf{z}_J\}$ to the query vectors. We ideally want to

$$\min_{\mathbf{z}_1, \ldots, \mathbf{z}_J} \max_{\mathbf{q} \in \mathcal{Q}, \mathbf{k} \in \mathcal{K}} \min_{j=1}^{J} \left| \mathbf{q}^T \mathbf{k} - \mathbf{z}_j^T \mathbf{k} \right|$$

or,

$$\min_{\mathbf{z}_1,...,\mathbf{z}_J} \max_{\mathbf{q} \in \mathcal{Q}, \mathbf{k} \in \mathcal{K}} \min_{j=1}^{J} \left( \mathbf{q}^T \mathbf{k} - \mathbf{z}_j^T \mathbf{k} \right)^2$$

$$\min_{\mathbf{z}_1,...,\mathbf{z}_J} \max_{\mathbf{q} \in \mathcal{Q}, \mathbf{k} \in \mathcal{K}} \min_{j=1}^{J} \left( \mathbf{q} - \mathbf{z}_j \right)^T \mathbf{k}\mathbf{k}^T \left( \mathbf{q} - \mathbf{z}_j \right)$$

Instead of minimizing the maximum over all possible $\mathbf{q}, \mathbf{k}$, we can minimize the sum or mean, i.e.,

$$\min_{\mathbf{z}_1,...,\mathbf{z}_J} \mathbb{E}_{\mathbf{q} \in \mathcal{Q}, \mathbf{k} \in \mathcal{K}} \min_{j=1}^{J} \left( \mathbf{q} - \mathbf{z}_j \right)^T \mathbf{k}\mathbf{k}^T \left( \mathbf{q} - \mathbf{z}_j \right)$$

$$\min_{\mathbf{z}_1,...,\mathbf{z}_J} \mathbb{E}_{\mathbf{q} \in \mathcal{Q}} \min_{j=1}^{J} \left( \mathbf{q} - \mathbf{z}_j \right)^T \left( \mathbb{E}_{\mathbf{k} \in \mathcal{K}} \mathbf{k}\mathbf{k}^T \right) \left( \mathbf{q} - \mathbf{z}_j \right)$$

$$\min_{\mathbf{z}_1,...,\mathbf{z}_J} \mathbb{E}_{\mathbf{q} \in \mathcal{Q}} \min_{j=1}^{J} \left( \mathbf{q} - \mathbf{z}_j \right)^T \left( \boldsymbol{\Sigma}_k + \boldsymbol{\mu}_k \boldsymbol{\mu}_k^T \right) \left( \mathbf{q} - \mathbf{z}_j \right)$$

Letting $\|\mathbf{x}\|_M^2 = \mathbf{x}^T \mathbf{M} \mathbf{x}$

$$\min_{\mathbf{z}_1,...,\mathbf{z}_J} \mathbb{E}_{\mathbf{q} \in \mathcal{Q}} \min_{j=1}^{J} \|\mathbf{q} - \mathbf{z}_j\|_{\boldsymbol{\Sigma}_k + \boldsymbol{\mu}_k \boldsymbol{\mu}_k^T}^2$$

which is same as the weighted K-means objective. The weights depend on the covariance $\boldsymbol{\Sigma}_k$ and mean $\boldsymbol{\mu}_k$ of the key vectors. Thus, in order to learn good K-means approximations of the query vectors from a discrete set, we introduce the following model.

In the `VQ-TR` model we modify the Transformer's encoder architecture by first mapping the $C$ incoming vectors, denoted by $X \in \mathbb{R}^{C \times F}$, through a VQ module:

$$Z_0, \text{indices} := \text{VQ}(X)$$

which will return the sequence of $C$ indices of the set of only $J$ vectors denoted by $Z_0 \in \mathbb{R}^{J \times F}$. We can apply Transformer based cross-attention to obtain latent $Z_1 \in \mathbb{R}^{J \times F}$:

$$Z_1 := \text{CrossAttn}(X, Z_0).$$

Since there are only $J$ latent vectors and typically in practice $J \ll C$, we can process them further via self-attention $L$ times:

$$Z_{l+1} := \text{SelfAttn}(Z_l).$$

Finally, we return the original number of sequence by gathering the resulting latent via the indices with respect to the quantization of the input vectors $X$:

$$\mathbb{R}^{C \times F} \ni Z := \text{Gather}(Z_{L+1}, \text{indices}).$$

This construction leads to an architecture with memory and compute complexity of $O(CJ) + O(LJ^2)$ from the the cross-attention and latent self-attention respectively (Jaegle et al., 2021; Hawthorne et al., 2022) per number of encoding layers $N$. One downside however to this architecture is that we lose the ability to model causal latents. On the other hand, as is commonly done in time series forecasting, we will use this non-causal encoder part of our architecture to learn discrete representations of large context windows, while the decoder will be the *causal* Transformer decoder and therefore scale $O(MP^2)$ for $M$ decoding layers. Since $P \ll C$ for the datasets we train on, this will not hinder us from training or doing inference conditioned on large histories. We present a schematic of the `VQ-TR` model in Figure 1 for both the training (Section 3.1) and inference (Section 3.2) scenarios.

A added benefit of this approach is that we can mitigate the quadratic memory and computation issue caused by a large sequence of input vectors in the vanilla Transformer to learn long term dependencies.

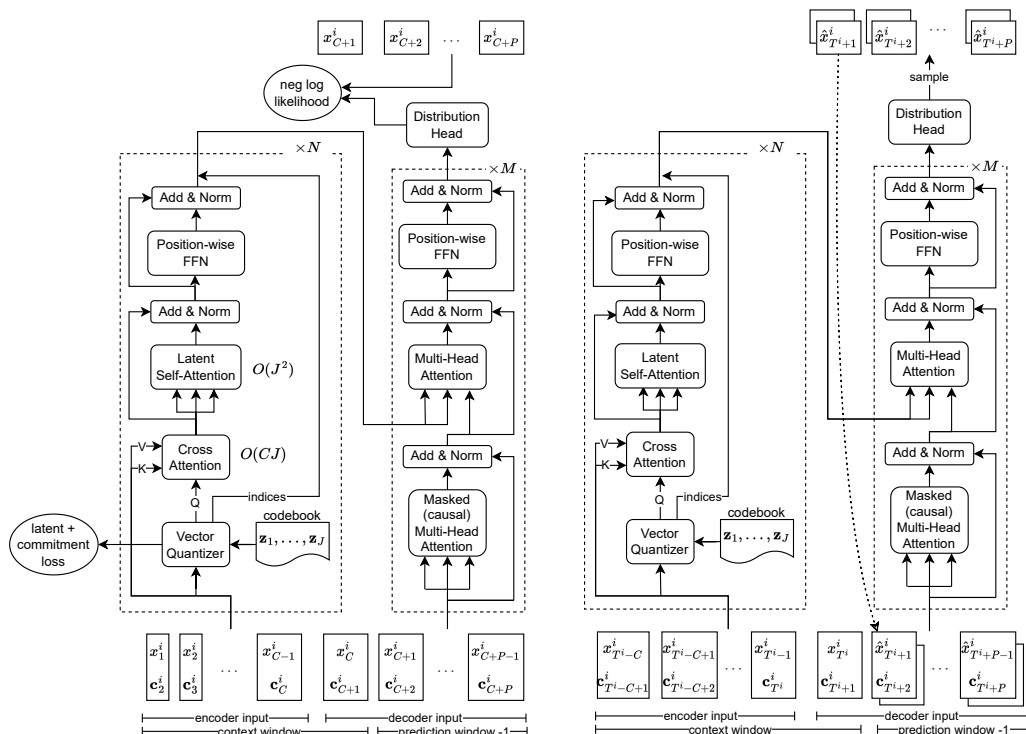

Figure 1: VQ-TR model which consists of $N$ encoding vector-quantized cross-attention blocks and $M$ *causal* decoding transformer blocks. During training (left) the encoder can take a potentially long sequence of length $C - 1$ from a time series and the decoder outputs the prediction length $P$ parameters of some chosen distribution which are learned via the negative log-likelihood together with the $N$ Vector Quantizer losses. During inference (right) we pass the last $C - 1$ length context window seen during training to the encoder and the very last value in training to the decoder, which allows us to sample the next time step which we can autoregressively pass back to the decoder to obtain predictions for our desired horizon.

## 3.1 TRAINING

Given a set $\mathcal{D}_{\text{train}}$ of $D \geq 1$ time series, we construct batches $\mathcal{B}$ of inputs by randomly sampling time series $\{x^i_{1:T^i}\}$, with $i \in \mathbb{Z}^+$ such that $i \leq D$, then selecting random $t \in \mathbb{Z}^+$ with $t \leq T^i - C - P$, and sampling context windows $\{x^i_{t:t+C}\}$, and subsequent prediction windows $\{x^i_{t+C:t+C+P}\}$, for fixed context window length $C$ and prediction window length $P$.

We can then for each batch step minimize the negative log-likelihood of the predicted distribution with respect to the ground truth predictions together with the $N$ latent and commitment losses from the VQ module of the encoder jointly. This is in contrast to the practice of first learning the discrete latent representations in an unsupervised fashion and then using these latents for down stream tasks as in for example the DALL·E (Ramesh et al., 2021) model.

## 3.2 INFERENCE

At inference time we go over each time series $i \in \mathcal{D}_{\text{train}}$ and feed the *last* context sized window (except for the last entry) to the encoder and the very last entry to decoder to obtain the parameters of the distribution of the next time point. We can now sample one or more values from this distribution and feed it back to the decoder to obtain samples for each time point of our desired horizon of $P$ time steps.

Note that we only need to run the encoder once in order to predict and can repeat tensors in the batch dimension to obtain many samples from the distribution in parallel. If a point forecast is required

then we can evaluate the empirical mean or median at each time point of the prediction. Unlike some generative modeling models, here we do not sample from a reduced temperature distribution to obtain high quality samples as we are interested in the empirical data distribution of the next time point conditioned on the past as well as covariates.

### 3.3 COVARIATES

Positional encoding give the Transformer the ability to encode positional information of sequences when needed since Attention is a permutation equivariant layer. In the time series setting we can naturally create positional encodings like Rotary Positional Embedding (RoPE) (Su et al., 2021) via date-time covariates. More specifically, for a particular time point $t$, depending on the frequency of the time series $i$, we can create hour-of-day, day-of-week, week-of-month, etc. features as a vector we denote by $\mathbf{c}_t^i$. Due to their temporal nature we can build these covariates for all future time points we wish to forecast for. Additional covariates can be constructed by considering the running means, the age of a time series as well as embedding the identity $i$ of each time series in a dataset via Embedding layers, as done in the `DeepAR` method.

### 3.4 SCALING

As detailed in the Salinas et al. (2019b), time series data can be of an arbitrary numerical magnitude within a dataset. This is unlike the vision, NLP or even audio modalities, and so in order to train a shared model over potentially very different time series we calculate the mean value of the signal within its context window and divide the signal with it to normalize it. This context window scale value is kept as a covariate and more importantly the model's output distribution is transformed back to the original scale via it during training and inference to calculate the log-probabilities or to sample from respectively. If the scaling cannot be done in the output distribution's parameter space one can also do it in the data space after sampling. *All* deep learning based methods in Section 4 incorporate this heuristic.

## 4 EXPERIMENTS

We test the performance of `VQ-TR` for the forecasting task in this section with respect to a number of methods on a number of open datasets.

Table 1: Number of time series, domain, frequency, total training time steps and prediction length properties of the training datasets used in the experiments.

| Dataset | $D$ | Dom. | Freq. | Time step | Pred. len. |
|---------|-----|------|-------|-----------|------------|
| Exchange | 8 | $\mathbb{R}^{\geq 0}$ | day | $6,071$ | 30 |
| Solar | 137 | $\mathbb{R}^{\geq 0}$ | hour | $7,009$ | 24 |
| Elec. | 320 | $\mathbb{R}^{\geq 0}$ | hour | $15,782$ | 24 |
| Traffic | 862 | $(0,1)$ | hour | $14,036$ | 24 |
| Taxi | $1,214$ | $\mathbb{N}^{\geq 0}$ | 30-min | $1,488$ | 24 |
| Wikipedia | $9,535$ | $\mathbb{N}^{\geq 0}$ | day | 762 | 30 |

For our experiments we use the following open datasets: `Exchange` (Lai et al., 2018), `Solar` (Lai et al., 2018), `Elec.`[1], `Traffic`[2], `Taxi`[3], and `Wikipedia`[4] preprocessed exactly as in Salinas et al. (2019a), with their properties listed in Table 1. As can be noted in the table, we do not need to normalize scales for the `Traffic` dataset. From the names of the datasets, we see that we cover a number of time series domains including finance, weather, energy, logistics and page-views

We will compare `VQ-TR` with the following deep learning baseline *probabilistic* univariate models

---

[1] `https://archive.ics.uci.edu/ml/datasets/ElectricityLoadDiagrams20112014`
[2] `https://archive.ics.uci.edu/ml/datasets/PEMS-SF`
[3] `https://www1.nyc.gov/site/tlc/about/tlc-trip-record-data.page`
[4] `https://github.com/mbohlkeschneider/gluon-ts/tree/mv_release/datasets`

Table 2: Forecasting metrics (lower is better) using: `SQF-RNN` with 50 knots, `ETS`, `MQCNN`, and `IQN-RNN`, `DeepAR`, `VQ-AR` and **`VQ-TR`** with Student-T (`-t`), Negative Binomial (`-nb`) or IQN (`-iqn`) emission heads, on the open datasets. The best metrics are highlighted in bold.

| Dataset | Method | CRPS | QL50 | QL90 | MSIS | NRMSE | sMAPE | MASE |
|---|---|---|---|---|---|---|---|---|
| Exchange | SQF-RNN-50 | 0.010 | 0.013 | 0.006 | **14.15** | 0.020 | 0.013 | 1.800 |
| | DeepAR-t | 0.012 | 0.016 | 0.007 | 69.29 | 0.022 | 0.030 | 9.980 |
| | ETS | 0.008 | **0.010** | 0.005 | 15.89 | 0.015 | **0.011** | **1.517** |
| | IQN-RNN | **0.007** | **0.010** | **0.004** | 17.37 | **0.014** | 0.013 | 3.041 |
| | MQCNN | 0.015 | 0.016 | 0.011 | 60.04 | 0.026 | 0.045 | 5.440 |
| | VQ-AR-t | 0.010 | 0.013 | 0.007 | 18.10 | 0.019 | 0.015 | 2.658 |
| | **VQ-TR-t** | 0.008 | 0.010 | 0.005 | 34.38 | 0.015 | 0.019 | 2.936 |
| Solar | SQF-RNN-50 | 0.330 | 0.431 | 0.175 | 5.65 | 0.929 | **1.342** | 1.004 |
| | DeepAR-t | 0.418 | 0.543 | 0.254 | 7.33 | 1.072 | 1.393 | 1.275 |
| | ETS | 0.646 | 0.661 | 0.383 | 18.55 | 1.112 | 1.546 | 1.938 |
| | IQN-RNN | 0.373 | 0.491 | 0.165 | 5.99 | 1.037 | 1.356 | 1.150 |
| | MQCNN | 0.928 | 0.960 | 1.535 | 73.58 | 1.920 | 1.838 | 2.248 |
| | VQ-AR-iqn | 0.320 | **0.414** | 0.174 | 5.64 | **0.885** | 1.346 | **0.969** |
| | **VQ-TR-iqn** | **0.317** | 0.435 | **0.153** | **4.60** | 0.909 | 1.346 | 1.021 |
| Elec. | SQF-RNN-50 | 0.078 | 0.097 | 0.044 | 8.66 | 0.632 | 0.144 | 1.051 |
| | DeepAR-t | 0.062 | 0.078 | 0.046 | 6.79 | 0.687 | 0.117 | 0.849 |
| | ETS | 0.076 | 0.100 | 0.050 | 9.99 | 0.838 | 0.156 | 1.247 |
| | IQN-RNN | 0.060 | 0.074 | 0.040 | 8.74 | 0.543 | 0.138 | 0.897 |
| | MQCNN | 0.129 | 0.148 | 0.132 | 30.54 | 1.230 | 0.240 | 2.000 |
| | VQ-AR-t | 0.054 | 0.068 | 0.036 | **5.88** | 0.653 | 0.107 | **0.717** |
| | **VQ-TR-t** | **0.050** | **0.063** | **0.033** | 6.29 | **0.495** | **0.104** | 0.744 |
| Traffic | SQF-RNN-50 | 0.153 | 0.186 | 0.117 | 8.40 | 0.401 | 0.243 | 0.76 |
| | DeepAR-t | 0.172 | 0.216 | 0.117 | 8.02 | 0.472 | 0.244 | 0.89 |
| | ETS | 0.373 | 0.386 | 0.287 | 17.67 | 0.647 | 0.489 | 1.543 |
| | IQN-RNN | 0.139 | 0.168 | 0.117 | 7.11 | 0.433 | 0.171 | 0.656 |
| | MQCNN | 1.220 | 0.563 | 2.005 | 116.69 | 0.723 | 0.636 | 2.712 |
| | VQ-AR-t | 0.138 | 0.164 | 0.113 | 7.79 | 0.409 | 0.185 | 0.641 |
| | **VQ-TR-t** | **0.110** | **0.130** | **0.093** | **6.91** | **0.392** | **0.137** | **0.500** |
| Taxi | SQF-RNN-50 | 0.286 | 0.362 | 0.188 | 5.53 | **0.570** | 0.609 | 0.741 |
| | DeepAR-nb | 0.299 | 0.379 | 0.203 | 5.44 | 0.610 | 0.582 | 0.771 |
| | ETS | 1.059 | 1.297 | 0.617 | 12.24 | 2.147 | 1.159 | 1.552 |
| | IQN-RNN | 0.295 | 0.370 | 0.201 | 6.51 | 0.583 | 0.629 | 0.758 |
| | MQCNN | 1.262 | 1.451 | 0.488 | 48.61 | 2.645 | 0.912 | 3.041 |
| | VQ-AR-nb | 0.286 | 0.362 | 0.193 | 5.43 | 0.572 | 0.570 | 0.741 |
| | **VQ-TR-t** | **0.281** | **0.357** | **0.184** | **5.19** | **0.570** | **0.561** | **0.729** |
| Wiki. | SQF-RNN-50 | 0.283 | 0.328 | 0.321 | 23.71 | 2.24 | 0.261 | 1.44 |
| | DeepAR-nb | 0.321 | 0.383 | 0.361 | 26.48 | 2.354 | 0.327 | 1.852 |
| | DeepAR-t | 0.235 | 0.27 | 0.267 | 23.77 | 2.15 | 0.219 | 1.295 |
| | ETS | 0.788 | 0.440 | 0.836 | 61.68 | 3.261 | 0.301 | 2.214 |
| | IQN-RNN | **0.221** | **0.254** | **0.251** | 21.78 | **2.102** | **0.193** | **1.214** |
| | MQCNN | 0.398 | 0.453 | 0.327 | 38.79 | 2.202 | 0.379 | 2.336 |
| | VQ-AR-iqn | 0.231 | 0.266 | 0.252 | 22.09 | 2.106 | 0.208 | 1.261 |
| | **VQ-TR-iqn** | 0.231 | 0.269 | 0.260 | **21.17** | 2.121 | 0.213 | 1.269 |

- `DeepAR` Salinas et al. (2019b): an RNN based probabilistic model which learns the parameters of some chosen distribution for the next time point;
- `MQCNN` Wen et al. (2017): a Convolutional Neural Network model which outputs chosen quantiles of the forecast upon which we regress the ground truth via Quantile loss;

- `SQF-RNN` Gasthaus et al. (2019): an RNN based non-parametric method which models the quantiles via linear splines and also regresses the Quantile loss;
- `IQN-RNN` Gouttes et al. (2021): combines an RNN model with an Implicit Quantile Network (IQN) Dabney et al. (2018) head to learn the distribution similar to `SQF-RNN`;
- `VQ-AR` Rasul et al. (2022): an RNN based encoder-decoder model which quantizes its input via a VQ;

as well as the classical `ETS` Hyndman & Khandakar (2008) which is an exponential smoothing method using weighted averages of past observations with exponentially decaying weights as the observations get older together with Gaussian *additive* errors (E) modeling trend (T) and seasonality (S) effects separately.

We follow the recommendations of the M4 competition Makridakis et al. (2020) for reporting forecasting performance metrics. In this regard, we report the mean scale interval score Gneiting & Raftery (2007) (MSIS[5]) for a 95% prediction interval, the 50-th and 90-th quantile percentile loss (QL50 and QL90 respectively), as well as the Continuous Ranked Probability Score (CRPS) Gneiting & Raftery (2007); Matheson & Winkler (1976) score. The CRPS is a *proper* scoring rule which measures the compatibility of a predicted cumulative distribution function (CDF) $F$ with the ground-truth samples $x$ as

$$\mathrm{CRPS}(F, x) = \int_{\mathbb{R}} (F(y) - \mathbb{I}\{x \leq y\})^2 \, \mathrm{d}y,$$

where $\mathbb{I}\{x \leq y\}$ is 1 if $x \leq y$ and 0 otherwise. We approximate the CDF via empirical samples at each time point and the final metric is averaged over the prediction horizon and time series of a dataset. The point-forecasting performance of models is measured by the normalized root mean square error (NRMSE), the mean absolute scaled error (MASE) Hyndman & Koehler (2006), and the symmetric mean absolute percentage error (sMAPE) Makridakis (1993). For pointwise metrics, we use sampled *medians* with the exception of NRMSE, where we take the *mean* over our prediction samples.

The results of our extensive experiments are detailed in Table 2. As can be seen `VQ-TR` performs competitively with respect to the methods compared, where the models have been trained using the hyperparameters from their respective papers using Student-T (`-t`), Negative Binomial (`-nb`) or Implicit Quantile Network (`-iqn`) emission heads. In particular for `VQ-TR` we can afford to use a larger context length of $C = 20 \times P$, where $P$ is the prediction horizon for each dataset, with the total number of encoder layers $N = 2$ and decoder layers $M = 6$. For training we use $J = 25$ codebook vectors, batch size of 256 for 20 epochs using the Adam (Kingma & Ba, 2015) optimizer with default parameters, and a learning rate of 0.001. At inference time we sample $S = 100$ times for each time point and feed these samples in *parallel* via the batch dimension autoregressively through the decoder to produce the reported metrics. The full source code will be made available after the review period.

## 5 RELATED WORKS

This method separates the length of the input sequence from the computation of the attention block by using a discrete set of latents. This strategy of reducing the computational cost is similar to the `Perceiver` (Hawthorne et al., 2022), `Set Transformer` (Lee et al., 2019), `Luna` (Ma et al., 2021) and `Compression Transformer` (Rae et al., 2020) models. `Perceiver-AR` is the closest related method, however it is a decoder only architecture and thus at inference time with many parallel samples for the probabilistic forecasting usecase, we have to run the cross-attention over a large context window for $P$ times causing both memory and computation bottlenecks. The use of VQ in sequential generative modeling has been explored in Audio/Speech setting Dhariwal et al. (2020); Zeghidour et al. (2021); Baevski et al. (2020) where typically a VQ-VAE is trained on the data and then a generative model trained on these learned latents separately.

The use of VQ in time series forecasting problem has been explored in the `VQ-AR` (Rasul et al., 2022) model however it uses an RNN to encode the history of a time series to a discrete latent which

---

[5]`http://www.unic.ac.cy/test/wp-content/uploads/sites/2/2018/09/`
`M4-Competitors-Guide.pdf`

is then used for the forecasting decoder RNN. In contrast this work incorporates the VQ module as part of an approximate Attention block to mitigate issues of large temporal contexts over which to forecast over.

## 6 SUMMARY AND DISCUSSION

We have presented `VQ-TR` a novel architecture which scales linearly with the encoder sequence size as a probabilistic forecasting model and demonstrated its performance against competitive models on a number of open datasets. `VQ-TR` reports good performance at test time and we can control the trade-off of computation and memory use by increasing or decreasing the number of discrete latents $J$.

As the reader might guess, this architecture can also work in other sequential modeling usecases and in future work we would like investigate the performance of `VQ-TR` for NLP, Audio or Vision based problems.

### ACKNOWLEDGMENTS

We acknowledge and thank the authors and contributors of the many open source libraries that were used in this work, in particular: GluonTS (Alexandrov et al., 2020), Vector Quantize PyTorch (Wang & Contributors, 2022), NumPy (Harris et al., 2020), Pandas (Pandas development team, 2020), Matplotlib (Hunter, 2007), Jupyter (Kluyver et al., 2016) and PyTorch (Paszke et al., 2019).

### ETHICS STATEMENT

Time series models considered in this work have been trained on open datasets which do not contain any personal information or ways of deducing personal information from them. All the models considered however can potentially be used, when trained with personal information, to predict behaviour or deduce private information like location for example, which could be potentially risky.

### REPRODUCIBILITY STATEMENT

We will open source the code to the paper after the review period. The dataset and preprocessing used in this work is based on open datasets with predefined preprocessing provided by GluonTS (Alexandrov et al., 2020). All the experiments have their hyperparameters in the appropriate Jupyter (Kluyver et al., 2016) notebooks.

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

## A  APPENDIX

### A.1  EXPERIMENTS ON TRANSFORMER BASED MODELS

We further compare against a range of Transformer based models and include the results in Table 3. We compare, apart from the vanilla `Transformer`, against:

- `TFT` Lim et al. (2021): an auto-regressive attention based Seq-to-Seq model with variable selection network for selecting relevant inputs;

- `Informer` Zhou et al. (2021): an efficient transformer and full horizon predictor model;

- `Autoformer` Wu et al. (2021): a transformer which decomposition the trend and seasonal components during the forecasting process together with series-wise auto-Correlation mechanism;

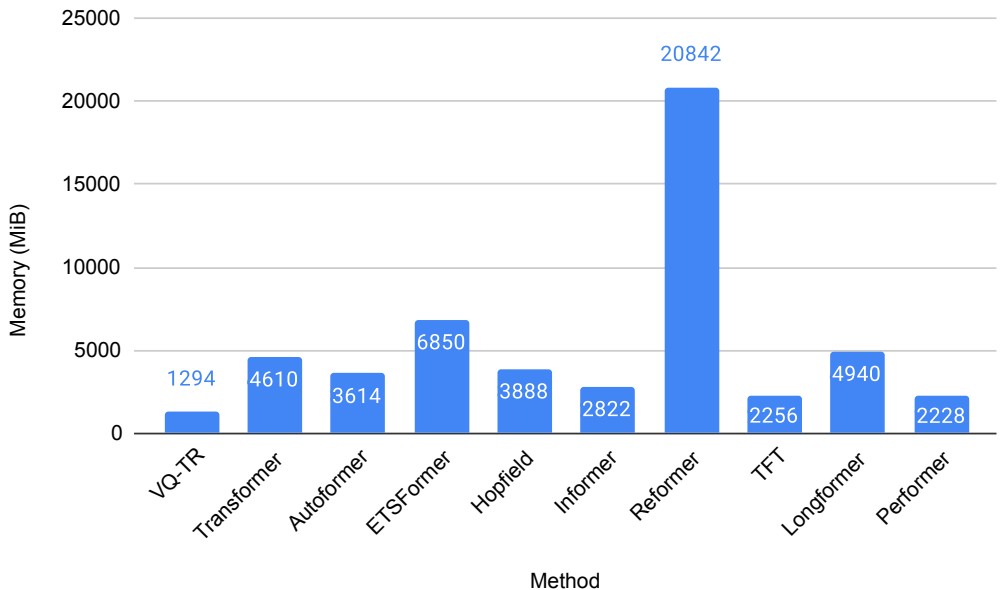

Figure 2: `VQ-TR` and other Transformer based model's memory usage when training on the different methods with the same hyper-parameters.

- `ETSformer` Woo et al. (2022): a transformer architecture which adds the principle of exponential smoothing and frequency attention in the attention mechanism;
- `Hopfield` Ramsauer et al. (2021): a modern Hopfield network with continuous state which generalizes attention;
- `Longformer` Beltagy et al. (2020): a local attention model with sliding window attention;
- `Performer` Choromanski et al. (2021): a transformer model which estimates regular full-rank attention by using linear space/compute complexity;

### A.1.1  MEMORY USE

To analyse the amount of memory for the different Transformer based methods, we plot this metric in Figure 2, when we train on `Traffic` using the same number of heads (2), encoding/decoding layers (2), feed-forward dimensions (16), context length (20 times prediction length) and batch size.

Table 3: Forecasting metrics (lower is better) using Vanilla `Transformer` and other Transformer based models with Student-T (`-t`), Negative Binomial (`-nb`) or IQN (`-iqn`) emission heads, on the open datasets. The best metrics are highlighted in bold.

| Dataset | Method | CRPS | QL50 | QL90 | MSIS | NRMSE | sMAPE | MASE |
|---|---|---|---|---|---|---|---|---|
| Exchange | Trans-t | 0.018 | 0.022 | 0.014 | 56.26 | 0.035 | 0.030 | 4.834 |
| | Tft-t | 0.064 | 0.072 | 0.086 | 1647.64 | 0.087 | 0.328 | 55.77 |
| | Informer-t | 0.012 | 0.015 | 0.006 | 28.89 | 0.024 | 0.020 | 2.779 |
| | Autoformer-t | 0.014 | 0.019 | 0.006 | 19.16 | 0.027 | 0.022 | 3.591 |
| | ETSformer-t | 0.009 | 0.013 | 0.006 | **13.51** | 0.019 | **0.014** | **2.148** |
| | Hopfield-t | 0.016 | 0.018 | 0.012 | 46.46 | 0.031 | 0.027 | 4.208 |
| | Reformer-t | 0.018 | 0.022 | 0.007 | 95.09 | 0.031 | 0.027 | 6.044 |
| | Linformer-t | 0.014 | 0.018 | 0.008 | 37.98 | 0.0266 | 0.020 | 2.822 |
| | Longformer-t | 0.021 | 0.025 | 0.009 | 57.34 | 0.044 | 0.028 | 3.810 |
| | Performer-t | 0.063 | 0.070 | 0.018 | 206.4 | 0.092 | 0.066 | 8.963 |
| | **VQ-TR-t** | **0.008** | **0.010** | **0.005** | 34.38 | **0.015** | 0.019 | 2.936 |
| Solar | Trans-t | 0.492 | 0.638 | 0.345 | 7.16 | 1.233 | 1.478 | 1.499 |
| | Tft-t | 0.931 | 0.995 | 1.305 | 48.04 | 2.03 | 1.950 | 1.950 |
| | Informer-t | 0.406 | 0.535 | 0.192 | 5.704 | 1.088 | 1.381 | 1.254 |
| | Autoformer-t | 0.758 | 0.985 | 0.308 | 15.68 | 2.035 | 1.854 | 2.317 |
| | ETSformer-t | 0.364 | 0.497 | 0.170 | 6.09 | 0.963 | 1.371 | 1.166 |
| | Hopfield-t | 0.477 | 0.642 | 0.243 | 5.94 | 1.217 | 1.471 | 1.505 |
| | Longformer-t | 0.432 | 0.560 | 0.211 | 6.41 | 1.122 | 1.411 | 1.314 |
| | Performer-t | 0.472 | 0.626 | 0.294 | 6.29 | 1.205 | 1.466 | 1.474 |
| | **VQ-TR-iqn** | **0.317** | **0.435** | **0.153** | **4.60** | **0.909** | **1.346** | **1.021** |
| Elec. | Trans-t | 0.061 | 0.078 | 0.035 | 7.49 | 0.538 | 0.115 | 0.853 |
| | Tft-t | **0.047** | **0.059** | **0.031** | **5.92** | 0.516 | **0.098** | **0.676** |
| | Informer-t | 0.064 | 0.079 | 0.054 | 6.47 | 0.739 | 0.116 | 0.788 |
| | Autoformer-t | 0.070 | 0.087 | 0.054 | 8.02 | 0.819 | 0.127 | 1.00 |
| | ETSformer-t | 0.068 | 0.081 | 0.064 | 8.43 | 0.650 | 0.128 | 0.904 |
| | Hopfield-t | 0.056 | 0.069 | 0.038 | 5.87 | 0.713 | 0.110 | 0.736 |
| | Reformer-t | 0.065 | 0.080 | 0.045 | 7.36 | 0.699 | 0.116 | 0.835 |
| | Linformer-t | 0.062 | 0.078 | 0.042 | 8.504 | 0.556 | 0.127 | 1.024 |
| | Longformer-t | 0.274 | 0.366 | 0.143 | 17.27 | 2.765 | 0.352 | 3.465 |
| | Performer-t | 0.163 | 0.202 | 0.119 | 20.20 | 1.326 | 0.248 | 2.470 |
| | **VQ-TR-t** | 0.050 | 0.063 | 0.033 | 6.29 | **0.495** | 0.104 | 0.744 |
| Traffic | Trans-t | 0.241 | 0.294 | 0.172 | 11.50 | 0.521 | 0.394 | 1.300 |
| | TFT-t | 0.139 | 0.165 | 0.108 | 7.82 | 0.425 | 0.213 | 0.648 |
| | Informer-t | 0.117 | 0.138 | 0.096 | 6.813 | 0.404 | 0.148 | 0.528 |
| | Autoformer-t | 0.184 | 0.225 | 0.146 | 9.33 | 0.500 | 0.272 | 0.901 |
| | ETSformer-t | 0.165 | 0.197 | 0.137 | 9.35 | 0.495 | 0.260 | 0.783 |
| | Hopfield-t | 0.118 | 0.140 | 0.095 | 6.68 | 0.406 | 0.142 | 0.534 |
| | Longformer-t | 0.317 | 0.382 | 0.278 | 15.92 | 0.694 | 0.556 | 1.651 |
| | Performer-t | 0.332 | 0.402 | 0.204 | 15.43 | 0.644 | 0.483 | 1.736 |
| | **VQ-TR-t** | **0.110** | **0.130** | **0.093** | **6.91** | **0.392** | **0.137** | **0.500** |
| Taxi | Trans-nb | 0.308 | 0.388 | 0.212 | 6.09 | 0.628 | 0.594 | 0.790 |
| | Tft-nb | 0.301 | 0.377 | 0.211 | 6.27 | 0.617 | 0.584 | 0.767 |
| | Informer-nb | 0.326 | 0.407 | 0.230 | 7.110 | 0.649 | 0.634 | 0.825 |
| | Autoformer-nb | 0.365 | 0.458 | 0.273 | 7.38 | 0.726 | 0.648 | 0.916 |
| | ETSformer-nb | 0.311 | 0.393 | 0.211 | 5.85 | 0.634 | 0.597 | 0.797 |
| | Hopfield-nb | 0.340 | 0.424 | 0.265 | 6.91 | 0.685 | 0.634 | 0.850 |
| | Linformer-t | 0.648 | 0.951 | 0.493 | 8.326 | 1.094 | 1.804 | 1.855 |
| | Longformer-nb | 0.398 | 0.473 | 0.320 | 7.40 | 0.652 | 0.905 | 0.937 |
| | Performer-nb | 0.397 | 0.471 | 0.297 | 7.18 | 0.626 | 0.954 | 0.954 |
| | **VQ-TR-t** | **0.281** | **0.357** | **0.184** | **5.19** | **0.570** | **0.561** | **0.729** |
| Wiki. | Trans-nb | 0.366 | 0.394 | 0.517 | 84.20 | 25.225 | 0.354 | 1.837 |
| | Tft-nb | 0.341 | 0.361 | 0.494 | 32.36 | 7.18 | 0.286 | 1.566 |
| | Informer-nb | 0.253 | 0.292 | 0.283 | 24.03 | 2.151 | 0.238 | 1.357 |
| | Hopfield-nb | 2.971 | 1.959 | 10.11 | 1630 | 124.8 | 0.456 | 8.367 |
| | Longformer-nb | 0.529 | 0.487 | 0.677 | 59.04 | 2.571 | 0.479 | 2.211 |
| | Performer-nb | 0.461 | 0.401 | 0.491 | 31.68 | 2.283 | 0.326 | 1.762 |
| | **VQ-TR-iqn** | **0.231** | **0.269** | **0.260** | **21.17** | **2.121** | **0.213** | **1.269** |

