# OpenReview forum: "VQ-TR: Vector Quantized Attention for Time Series Forecasting"
_ICLR.cc/2023/Conference — Submitted to ICLR 2023_

### Official Review · Reviewer_TMte · 2022-10-24

**Confidence:** 3
**Correctness:** 3
**Technical Novelty And Significance:** 3
**Empirical Novelty And Significance:** 3
**Recommendation:** 5

**Clarity, Quality, Novelty And Reproducibility:**

The paper is generally well written and clear. However, for full reproducitbility/evaluation additional details on training would be required -- specifically with regards to how feature engineering and hyperparameter optimisation are performed.

**Strength And Weaknesses:**

Strengths
---

While transformer models have demonstrated strong performance on time-series datasets, the slow performance of the attention mechanism have made applications difficult in low-latency settings. Methods to improve performance are important, and vector quantisation is an interesting approach. The paper is also fairly well-written and motivated, and the arguments are clearly lined out.

Weaknesses
---

While the approach is promising, I do have several key concerns:

1.	Results for the model are rather weak, with outperformance demonstrated on 4 out of the 6 dataset only when compared to fairly simple neural network baselines. Are there any specific dataset characteristics that cause the approach to underperform?

2.	How does the model compare against other transformer approaches? The true test of the model should be against methods with the full attention mechanism to evaluate how much performance is sacrificed for efficiency. How does the model stack up against the Pyraformer (ICLR 2022) which also has linear time/space complexity?


**Summary Of The Paper:**

The paper proposes a new efficient transformer model for time series forecasting – which mitigates the typical O(N^2) forecast time using a approximation based on vector quantisation. The model demonstrates outperformance over other O(N) RNN methods across a variety of datasets.

**Summary Of The Review:**

On the whole the approach does appear to hold promises, although additional comparable benchmarks are required to fully demonstrate the value proposition of the approximation.

---

### Official Review · Reviewer_PERd · 2022-10-24

**Confidence:** 3
**Correctness:** 3
**Technical Novelty And Significance:** 3
**Empirical Novelty And Significance:** 2
**Recommendation:** 3

**Clarity, Quality, Novelty And Reproducibility:**

This paper was written in good clarity. The introduction of vector quantization and `VQ-TR` method was very detailed, figure 1 is very informative for understanding model architecture. Training & inference details are also covered. On the other hand, some details in the Experiment section seem a bit unclear. The point of proposing this method is to achieve linear complexity w.r.t sequence length in the Transformer model but whether this benefit is achieved or useful remains unclear, which impacts the quality of this paper. On the originality side, this paper followed the setting of probabilistic time series forecasting, and the concept of vector quantization was already used in `VQ-AR`. The creative part lies in the design of VQ based attention mechanism and how that fits in Transformer, which is still interesting and novel.


**Strength And Weaknesses:**

Overall, this paper follows up on the `VQ-AR` model and further incorporates the vector quantization module as part of the approximate attention block.

Strengths of this paper:
1. The presentation of `VQ-TR` model is clear, with rich details in demonstrating how it approximates the attention mechanism and how the linear complexity is derived.
2. Figure 1 was well generated and informative for understanding this model.
3. Experiments are extensive, covering multiple domains of time series and different baselines.
4. The idea seems generally applicable to more applications in the sequence representation learning area.

Weaknesses and how to improve:
1. The paper claimed to improve the theoretical complexity of Transformer's attention mechanism. This should potentially bring benefit from using a longer input length and outperforming Transformer based models with that. More experiments with long input length against vanilla Transformer based models would make this point more valid.
2. Similar to the previous point, the improvement over theoretical complexity can be quantified by benchmarking against the vanilla Transformer model. Would be great to see numbers on how it improves inference speed or memory occupation.
3. The performance seems competitive but there's a lack of analysis of why the model doesn't work well for some of the datasets. Some case study or analysis would fix this problem.
4. The hyper-parameter `J` of `VQ-TR` model was claimed to control the `trade-off of computation and memory use`, and would like to see some ablation study on how this hyper parameter is selected and the actual impact on performance.
5. In the Experiment section, the use of different emission heads is a bit confusing. Can you please add some explanation on how the emission head is selected for certain tasks, especially when `VQ-AQ` and `VQ-TR` use different ones?


**Summary Of The Paper:**

This paper introduced a `VQ-TR` method that maps larger sequences to a discrete set of latent representations to be used as part of the attention module. In this way, larger context windows can be attended with linear complexity w.r.t sequence length. Experiments show that this method performs competitively against the baseline methods.

The contributions of the paper come from incorporation of ths vector quantization attention module in Transformer to get linear computation and memory use w.r.t. sequence size, theoretically analyze how the vector quantization method approximated the vanilla attention mechanism, and demonstrate competitive its performance.


**Summary Of The Review:**

To summarize, this paper proposed this `VQ-TR` design that uses vector quantization to approximate the attention mechanism in Transformer, and utilized that for probabilistic time series forecasting. The paper presented its model design clearly and provided details for theorem proof for the usage of vector quantization in attention. On the other hand, even though the experiments demonstrated the method is competitive, it does not take advantage of the complexity advantage of linear complexity and show how that could be leveraged for better prediction and how that improves efficiency in a quantified way. Therefore, I recommend this paper to be rejected.

---

> ### Author Response · Authors · 2022-11-18
> **Ablation**
>
> Dear Reviewer,
>
> Thank you for your kind review. As mentioned above we have added an extensive table comparing a range of other transformer based models as well as a table of memory usage.
>
> We have also mentioned below potentially why the model does not work well on two datasets. When we mean not well it is still 2nd best or close to the best metrics.
>
> We have started to run an ablation of the size J, and will add it to the pdf. Our initial results suggest that the size of J depends on the datasets we train on and so will need to run further experiments.
>
> However, we can also think of the number of J as a way to reduce the computation costs like in the perciever-AR paper.
>
> For real-valued datasets we select a continuous distribution head or IQN head and for count valued data we select the negative binomila head.
>
> I hope this clears up your confusion and concerns?
>
> Thank you!

---

### Official Review · Reviewer_miFG · 2022-10-25

**Confidence:** 3
**Correctness:** 3
**Technical Novelty And Significance:** 3
**Empirical Novelty And Significance:** 3
**Recommendation:** 6

**Clarity, Quality, Novelty And Reproducibility:**

The paper is somewhat difficult to follow, but it seems that the idea is novel and potentially can inspire many future studies. For reproducibility, some operations (latent self-attention, latent cross-attention, etc.) could be more explained in detail. Although the VQ process takes an important part of the paper, there is not much explanation (i.e., VQ training loss, VQ regularization, hyper-parameters, etc.). I hope the code release can solve these.

**Details Of Ethics Concerns:**

The authors addressed potential concerns related to the paper.

**Strength And Weaknesses:**

Strengths:
- As I know, this is the first work to compute cross-attention and self-attention between VQ codewords. This approach greatly reduces the computation cost, which is the main obstacle to long-range prediction.
- The paper properly compared previous competitors for various benchmarks. Also, the performance seems good enough.

Weaknesses:
- Although the authors mention that Transformers “scale poorly to the size of the sequence length”, there is no empirical evidence to support this claim.
- Are there other metrics that are related to computation cost/inference time/parameter size, etc.? I am not sure that the compared models have experimented using a similar budget (i.e., fair comparison).
- Just a suggestion; There exist some Transformer models that incorporate (K-means) clustering for query/key computation. Although not directly related, comparing those works can be helpful.

[1] Fast Transformers with Clustered Attention (NeurIPS 2020)

[2] Efficient Content-based Sparse Attention with Routing Transformers (TACL 2020)

**Summary Of The Paper:**

The paper proposes a novel Transformer architecture for extremely long-range time series forecasting. The key component of the proposed method is VQ, where the number of possible output features is limited. As a result, the computation only happens between quantized vectors, and the heavy quadratic cost of self-attention can be greatly reduced. Experimental results show that the proposed VQ-TR achieves comparable and sometimes better performance than other competitors.

**Summary Of The Review:**

Overall, the paper tackles an important problem. Using VQ for computation reduction itself may not be new, but the realization/implementation is novel enough. However, some parts can be improved (see weaknesses above).

---

### Official Review · Reviewer_ZX4K · 2022-10-31

**Confidence:** 4
**Correctness:** 1
**Technical Novelty And Significance:** 3
**Empirical Novelty And Significance:** 2
**Recommendation:** 1

**Clarity, Quality, Novelty And Reproducibility:**

Clarity and quality should be improved - as mentioned above.  The idea does seem interesting and novel (at least in its application to time series forecasting - as the idea is borrowed from an existing approach for other applications of transformers to temporal data).  The authors state they will release the code which will help with reproducibility, but otherwise the method would need to be described with greater detail, preciseness and clarity to enable reproducing.

**Strength And Weaknesses:**

Strengths:

1. The quantization idea is interesting and I don't believe has been applied to transformer forecasting before.

2. The authors provide theoretical analyses on approximation errors to help motivate the idea.

3. Extensive experiments are performed to compare forecast error using point and probabilistic forecast metrics on multiple datasets.


Weaknesses:

1. The biggest weakness is that the method is motivated as a way to speed up transformers for forecasting for long context windows.  However, surprisingly, no transformer forecast methods were compared with, and no analyses or comparison or results of any kind are reported for runtimes and how it varies with context length.  Ideally the recent state of the art transformer forecast methods should be compared with (e.g., Fedformer, Informer, etc.) both in terms of accuracy, but also in terms of run times, especially for varying context length.  At the very least, showing the runtime variation of the basic transformer forecasting approach vs the proposed one should be provided in experiment results, as context and forecast horizon length vary.  It's also important to show how forecast error changes with varying context length - to validate the motivating claim that longer context window length improves forecasts and thus speeding up this part of the model is important.


2. Furthermore, there are a lot of techniques for speeding up transformers, and many have been applied to transformer forecasting as well - these should be compared to (and be part of related work too to set the context).  For example, in "Long-Range Transformers for Dynamic Spatiotemporal Forecasting" they use a Performer style approach to enable scaling the attention - which uses a linear approximation to the attention using a random kernel approach.


3. The method only addresses speeding up the calculation of the encoder part of the forecast model - which seems insufficient to enable general scalability - especially since in practice the forecast horizon can be just as long as the context window (or even longer).  I.e., in Section 3 - it is stated "Since P << C for the datasets we train on..." - but C is essentially a modeling choice or hyper parameter (not a feature of the dataset) and it seems rarely the case for it to be chosen much larger than P.   I.e., it is often not the case - in fact a lot of forecasting work in the literature has found using smaller context windows to work best in many cases (that could be even less than the forecast horizon) as long as appropriate features / exogenous series are used.


4. The model formulation is not provided with sufficient detail and clarity - many specifics and even variables used in equations are not defined or explained.  Overall it feels like an incomplete, "hand-wavy" description of the approach and the model.  Some specific comments / examples:
    -  In 2.1, C used in multiple equations is never defined, and it should be defined.
    - Also here the description is confusing and not very clear or precise - making it hard to follow.  E.g., in 2.1 it's stated "Afterwards M layers of a causal or masked decoding Transformer can be used to model the P future time points as:" but the following equation does not at all show how M layers are applied and doesn't even include M or any coefficient related to it.  Right after this the output model is introduced with "For example...", when it has nothing to do with the preceding sentence and the output model and this equation are not fully explained, again along with nomenclature used not being explained, such as the "N" being used but not defined or explained.
    - Section 3 is not explained clearly - shorthand operations are given without any complete and clear description of what they translate to (either mathematically or via textual description).  I would suggest moving the proof to the appendix (and adding more detail as they are difficult to follow), and possibly the formulation of the optimization objective following that, which is also not clearly explained.  In this way, more space would be made for providing full mathematical details of the model.


5.  There are some unsubstantiated statements / claims made - these should supported with references or sufficient arguments (with explanation that they are just opinions are hypotheses).
    - "Transformers offer good inductive bias for the forecasting task..." - this seems like a strong and unsubstantiated statement - is there some reference to back this claim up, or anything beyond a heuristic argument?
    - "...and consider some fixed sized context window sampled randomly from the full time series history to learn some historical representation and use this representation in the decoder to learn the distribution of the next time point..." - I disagree - the majority of approaches use a fixed sized context window not sampled randomly, but immediately preceding the current time points / the future time window being predicted.  For instance, if one is predicting t+1 through t+h, we use a context time window from t-c to t - not some random window from the history.


6. The experiment comparison may not be fair.  In particular, in section 3.4 the authors describe a window scaling approach they use for their method.  However, a similar window scaling approach was shown to be effective in ICLR 2021 paper "Reversible Instance Normalization for Accurate Time-Series Forecasting against Distribution Shift" across a wide range of models.  I.e., when applied to a wide variety of different models it significantly improved their forecast accuracy across multiple datasets (I believe the same set used here).  However, only the proposed model here uses the window-scaling normalization approach - so it's hard to say if the other methods wouldn't work as well or better if also using the window scaling - i.e., any observed benefit may just be from using window scaling.
Additionally particulars of how all configurations / hyper parameters are chosen should be given - in particular in the reported results - why are only some output distributions shown for some datasets - how were these selected - and why weren't results shown for all.
Finally, std. dev. should also be reported over multiple random runs and shifted test windows.



Minor:
In section 2 you use the term "data-time" without definition - a term which I have never seen before.  Is this perhaps meant to be "date-time" (i.e., a combination of data and time and commonly used in computing and libraries for time series and for referring to particular point in time)?

**Summary Of The Paper:**

The authors propose an approach to speed up transformer based forecasting / enable it to scale better for longer context (history) windows feeding into the transformer forecasting model.  Typically in space and time complexity transformer forecasting would scale quadratically with this context window (sequence) length.  Here they propose to adopt a quantization approach used elsewhere for the encoder part of the forecast model, which encodes each input in the sequence to a finite dimension codebook, applies cross-attention on the encoded vectors, and then applies self attention on the reduced set of vectors to transform them before mapping back to the input sequence positions - in this way enabling linear scaling with context window length, for deriving the output of the encoder portion of the forecast model.  The decoder part of the model is left to handle the causal temporal modeling with the regular self attention approach across the full output sequence (so it is quadratic in the forecast horizon).

The authors compare the proposed approach with different output distributions on a number of datasets to a variety of methods, showing competitive performance.

**Summary Of The Review:**

This work has potential, but overall feels like a work in progress - a lot more is left to be done before it is ready for publication in a top conference.  This includes the experiments needed to validate the main claims and motivations for the proposed method, which are currently lacking.

---

### Author Response · Authors · 2022-12-07
**Reminder**

Dear Reviewers,

We wanted to kindly remind you that the review period is ending soon and would like to ask you to join the discussion here. We hope that our responses and updated benchmarks have cleared up the main points raised by you all.

If you have further questions please ask them here.

Thank you!

---

### Decision · Program_Chairs · 2023-01-20

**Decision:**

Reject

**Justification For Why Not Higher Score:**

This paper totally misses the transformer baselines in its first version. Although additional experiments are included in the rebuttal period, they are still generally incomplete, and more baselines and relevant ablation studies are needed.

**Justification For Why Not Lower Score:**

N/A

**Metareview: Summary, Strengths And Weaknesses:**

This paper proposes an efficient transformer VQ-TR for long-time series forecasting. By mapping the time series to a discrete set of latent variables, VQ-TR can reduce the time complexity of attending large context windows to be linear in the sequence length. Both theoretical analysis and empirical results are provided.

Improving the scalability of transformer-based time series models is a meaningful and important problem, and all reviewers agreed that VQ-TR is quite novel and generally applicable. However, the reviewers had major concerns about the total missing transformer baselines in the evaluation (Reviewers ZX4K, miFG, PREd and TMte) and the lack of ablation studies (Reviewers ZX4K and PREd). Some reviewers (Reviewers ZX4K and miFG) also criticized the writing of the paper. Although the authors gave efforts to address some of those concerns, several main concerns are still not fully addressed, including the missing SOTA transformer baselines, the unfinished ablation study, and some unsupported claims pointed out by Reviewer ZX4K. Thus, the current version of this paper has not met the criteria for publication in ICLR.